# A KAN-based hybrid deep neural networks for accurate identification of transcription factor binding sites

Guodong He, Jiahao Ye, Huijun Hao, Wei Chen *

School of Information Engineering, Wenzhou Business College, Wenzhou, Zhejiang, PR China

* chenw@wzbc.edu.cn

## Abstract

### Background

Predicting protein-DNA binding sites in vivo is a challenging but urgent task in many fields such as drug design and development. Most promoters contain many transcription factor (TF) binding sites, yet only a few have been identified through time-consuming biochemical experiments. To address this challenge, numerous computational approaches have been proposed to predict TF binding sites from DNA sequences. However, current deep learning methods often face issues such as gradient vanishing as the model depth increases, leading to suboptimal feature extraction.

### Results

We propose a model called CBR-KAN (where C represents Convolutional Neural Network (CNN), B represents Bidirectional Long Short Term Memory (BiLSTM), and R represents Residual Mechanism) to predict transcription factor binding sites. Specifically, we designed a multi-scale convolution module (ConvBlock1, 2, 3) combined with BiLSTM network, introduced KAN network to replace traditional multilayer perceptron, and promoted model optimization through residual connections. Testing on 50 common ChIP seq benchmark datasets shows that CBR-KAN outperforms other state-of-the-art methods such as DeepBind, DanQ, DeepD2V, and DeepSEA in predicting TF binding sites.

### Conclusions

The CBR-KAN model significantly improves prediction accuracy for transcription factor binding sites by effectively integrating multiple neural network architectures and mechanisms. This approach not only enhances feature extraction but also stabilizes training and boosts generalization capabilities. The promising results on multiple key performance indicators demonstrate the potential of CBR-KAN in bioinformatics applications.

**Data availability statement:** CBR-KAN is an open-source collaborative initiative available in the GitHub repository (https://github.com/QAQ1551QAQ/DNA-TF-binding-sites-predict.git).

**Funding:** This work was partly supported by the Research Project RC201911, Wenzhou Business College, China

**Competing interests:** The authors have declared that no competing interests exist.

**Abbreviations:** TF, Transcription Factor; ChIP-Seq, Chromatin Immunoprecipitation Sequencing; DNNs, Deep Neural Networks; CNN, Convolutional Neural Network; BiLSTM, Bidirectional Long Short-Term Memory; NLP, Natural Language Processing; LSTM, Long Short-Term Memory; KAN, Kolmogorov-Arnold Network; MLPs, Multi-Layer Perceptrons; TPR, True Positive Rate; FPR, False Positive Rate

## Introduction

Research on the function of DNA in the human genome is a complex and challenging task. Because the function of most non-coding DNA is still poorly understood, researching it is particularly important and has significant implications for biological research. A transcription factor (TF) is a type of protein that regulates gene expression by binding to upstream regulatory elements located in the promoter and enhancer regions of DNA, controlling the expression with specific intensity, timing, and spatiality. Identifying transcription factor binding sites is crucial for understanding transcriptional regulatory mechanisms [1].

Recent advances in next-generation sequencing have led to the development of Chromatin Immunoprecipitation Sequencing (ChIP-Seq) has been developed to identify protein-DNA binding sites [2]. ChIP-seq facilitates the discovery of sequence patterns and greatly improves the resolution of protein-DNA interactions. Although the number of identified protein-DNA binding sites has rapidly increased, the DNA sequences obtained from ChIP-Seq are still highly noisy [3]. Furthermore, ChIP-Seq analysis typically requires a large number of cell samples, which might limit its application in research. Therefore, computational methods for predicting protein-DNA binding sites need to be developed. At present, the developed methods can be roughly divided into traditional algorithms and deep learning algorithms, and deep learning algorithms have attracted more attention [4–6].

The recent abundance of high-throughput genome sequencing data has prompted the development of novel bioinformatics algorithms that can be used for analyzing large and rapidly increasing data sets [7]. Deep learning algorithms effectively address these issues as they can handle large data sets and efficiently identify complex patterns within feature-rich data [8]. Deep learning algorithms rely on extensive training data and specialized hardware to effectively train deep neural networks (DNNs) effectively. As a result, DNNs have been widely used to solve genomic problems such as motif discovery, gene expression inference, splicing site prediction, and regulatory element prediction [9–15].

Recent studies have employed DNNs to analyze large-scale chromatin profiling data and investigate their impact on TF binding function. DeepSEA constructed a CNN-based model to predict the impact of non-coding variations across multiple cell types, focusing only on capturing regulatory motifs to understand tissue-specific functions [16]. DanQ proposed a hybrid convolutional and recurrent deep neural network to quantify DNA functions, which captures both regulatory motifs and the regulatory grammar [17].

However, the performance of deep learning models can still be significantly improved. Most methods combine CNN and RNN to construct complex models, but encounter gradient vanishing problems as the models deepen, causing training difficulties. He *et al.* proposed a residual learning framework to train the models of networks which are significantly deeper than those used previously [18]. Meanwhile, an important progress of DNN is the attention mechanism inspired by the signal processing mechanism of human brain [19]. Deng et al. proposed a hybrid deep learning framework, termed DeepD2V, which combines recurrent and convolutional

neural networks to predict transcription factor binding sites, demonstrating superior performance and robustness on public ChIP-seq benchmark datasets [20]. It can quickly filter valuable information from vast amounts of raw data and address long-term dependency issues in RNNs.

To further improve the model performance, we introduced the KAN network, which replaces the traditional multi-layer perceptron and plays a vital role in improving the model robustness and feature extraction ability [21]. Through residual connections, we avoid the gradient vanishing problem and achieve better feature representation. Therefore, we have developed a more effective model for predicting protein-DNA binding sites by integrating the strengths of CNNs, BiLSTMs, KAN networks, and residual learning.

This study proposes a hybrid deep neural network based on KAN, named CBR-KAN, which is an advanced model for predicting transcription factor binding sites (TFBS) in DNA sequences. The model innovatively integrates multiple neural network architectures: it introduces Kolmogorov-Arnold Networks (KAN) into the TFBS prediction field for the first time, replacing traditional MLP structures, and enhances feature fusion capabilities through learnable spline basis functions. It incorporates Convolutional Neural Networks (CNN) for spatial feature extraction and employs Bidirectional Long Short-Term Memory Networks (BiLSTM) to capture sequence dependencies. Additionally, by introducing residual connections, the model ensures performance stability while deepening its structure. Overall, this hybrid strategy aims to significantly improve prediction accuracy by synergistically utilizing diverse neural network components tailored for bioinformatics.

## Materials and methods

### Dataset and processing

In this study, we used 50 public ChIP-seq data including GM12878, H1-hESC and K562 to evaluate the effect of the proposed model, These datasets are derived from three types of cell lines, including Gm12878, H1-hESC, and K562. For each cell line, approximately 15,000 top-ranked sequences are selected as positive samples from each record in the peak file, and each sequence consists of 200 base pairs. Negative samples are generated by matching the repeat fraction, length, and GC content of the positive samples following the method of Deng et al. [20]. It is worth noting that the preparation methods of ChIP-seq data and the training set have been adopted by DeepBind, DanQ, WSCNN, and WSCNNLSTM [20].

First, each input DNA sequence was segmented and then converted into a corresponding vector using pretrained dna-2vec vectors. Word embedding is widely used in natural language processing (NLP) to effectively map words into high-dimensional spaces using fixed-length vectors [22]. This concept has also been applied to DNA sequences [23]. In this study, we utilized pre-trained word vectors for embedding DNA sequences. We segment a DNA sample of length n (In this study, the length of n is 200) by means of window size m (m = 3) and step size s (s = 1), then we get n-2(i.e., the length is 198) DNA sequences of length m $x_i \in \{x_1, x_2, x_3, ..., x_{n-2}\}$. Each $x_i$ can be found in a pre-trained DNA vector matrix derived from dna2vec [24]. We use $e_i \in R^k$ to express the k (k = 100) dimensional vector of the slit i sequence, then our sequence $x_i$ can be transformed into $e_i \in \{e_1, e_2, e_3, ..., e_{n-2}\}$. Finally, for each sample of length n, it can be embedded as:

$$e_{1:n-2} = e_1 \oplus e_2 \oplus \cdots \oplus e_{n-2} \tag{1}$$

Where $\oplus$ represents the concatenation operator.

### Model building

The structure of the CBR-KAN model proposed in this paper is shown in Fig 1 and consists of four parts. We regard a DNA sequence as a sentence and the segmented fragments as the words that make up the sentence. Our model can be explained as follows: (1) Embedding layer: Map each word to a low-dimensional vector. (2) CNN layer: Use one-dimensional convolution and maximum pooling to obtain feature maps from the embedding layer. (3) Long short-term

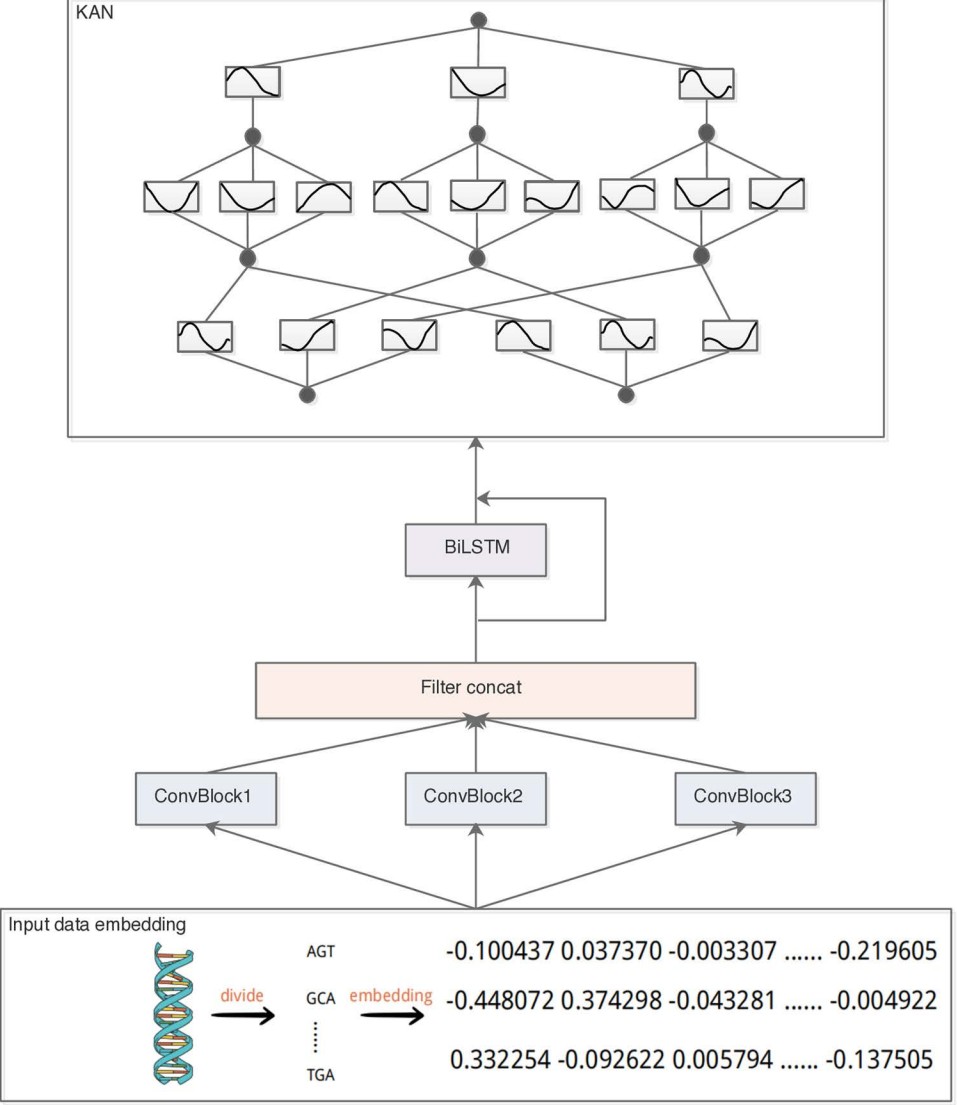

**Fig 1. A graphical illustration of the CBR-KAN model.** The structure and parameters of ConvBlock1, ConvBlock2, ConvBlock3 are shown in Fig 2.

memory (LSTM) layer: Use BiLSTM to obtain high-level features from the CNN layer. (4) KAN layer: Receiving features from CNN and BiLSTM, which capture the hierarchical structure and long-term dependency information of the data respectively, these features are effectively fused together through residual connections, allowing the KAN network to utilize both features for classification at the same time. In addition, dropout is used to prevent overfitting, and normalization is used to ensure the stability of the data feature distribution and accelerate the convergence of the model.

(1) CNN Layer

A growing number of researchers are trying to apply convolutional neural network to sequence data, where CNN models can shorten the sequences while extracting features since TextCNN was proposed in 2014 [25]. In this study, a kernel $w \in R^{hk}$ (h<=n-2, k = 100) is used to perform a one-dimensional convolution operation on the embedded DNA sample. The model uses a step size of 1 without padding. The generated feature $z_i$ can be represented as:

$$z_i = \int (w \cdot e_{i:n-h-1} + b)$$

<div align="right">(2)</div>

Where f is a nonlinear activation function, b is a biasing function and $z_i \in \{z_1, z_2, z_3, ..., z_{n-h-1}\}$. Meanwhile, we use maximum pooling for $z_i$ extraction, which can be represented as: $\hat{z}_i = max\{z_i\}$, in which $\hat{z}_i$ is the result after max pooling [26].

The structure and parameters of the CNN layer are shown in Fig 2. This module is composed of three convolutional blocks (ConvBlock1, ConvBlock2, ConvBlock3). Each block processes the embedded input data through a series of convolutional and pooling layers. ConvBlock1 consists of four convolutional layers with 160 output channels in the first three layers and 256 in the fourth layer, using kernel sizes of 9, 1, 5, and 8, each followed by ReLU activation and max pooling. ConvBlock2 and ConvBlock3 each contain two convolutional layers, increasing output channels from 128 and 180–256, with kernel sizes changing from 11 and 1–9 and 8, respectively, also followed by ReLU activation and max

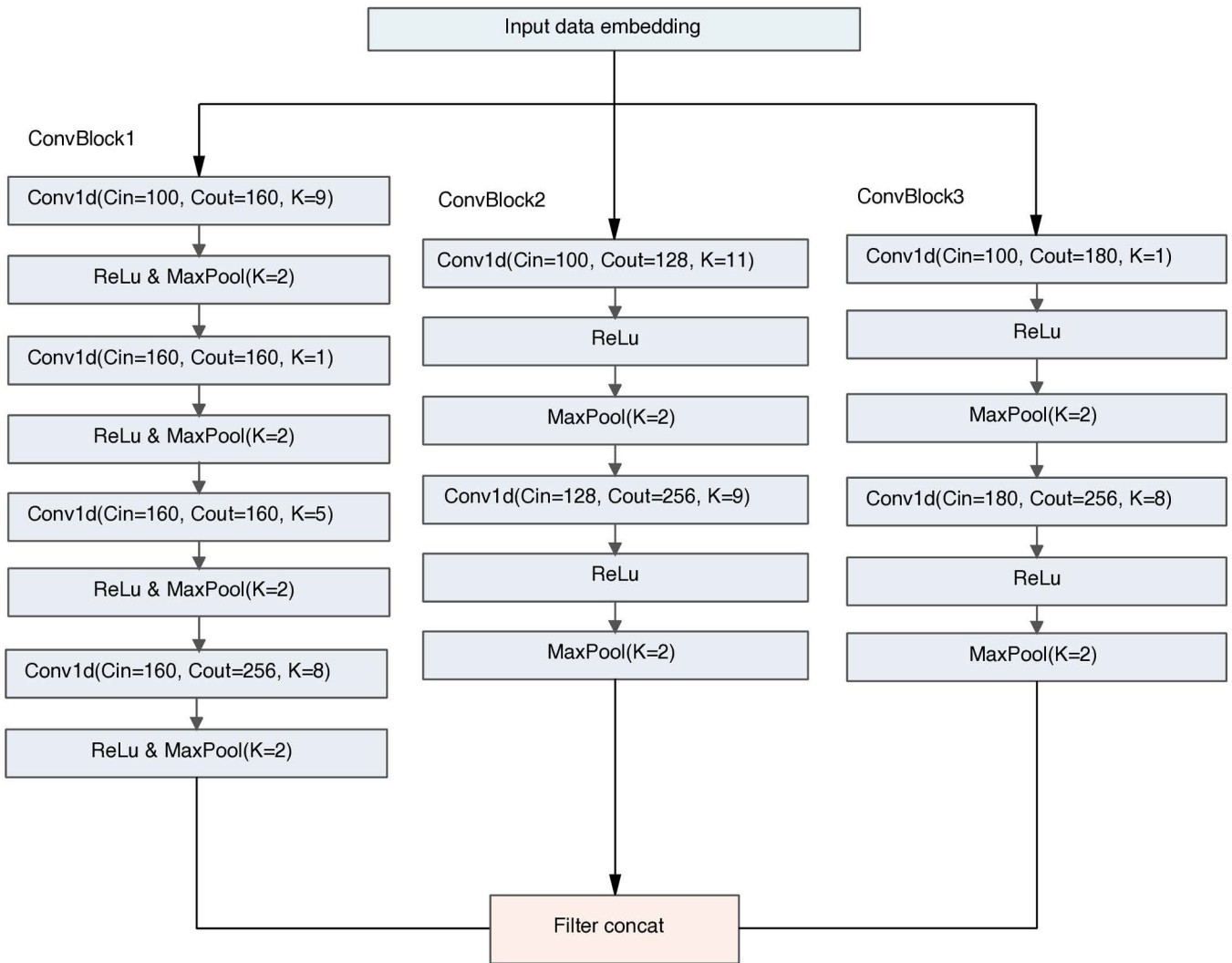

**Fig 2. The structure and parameters of CNN layer.** Conv1d: 1D convolution, Cin: Number of input channels, Cout: Number of channels produced by the convolution, K: Size of the kernel.

pooling. The outputs from these three blocks are concatenated into a single vector. The outputs of these three blocks are concatenated into a vector, and this architecture can capture multi-scale features and enhance the classification performance of the model.

(2) LSTM layer

LSTM was first proposed by Hochreiter and Schmidhuber in 1997 to overcome the gradient vanishing problem [27]. Graves *et al*. introduced a variant of LSTM called BiLSTM, the specific formula for the BiLSTM unit corresponding to the input $x_t$ at position t is as follows:

$$i_t = \sigma(W_{xi}x_t + W_{hi}h_{t-1} + W_{ci}c_{t-1} + b_i)$$

(3)

$$f_t = \sigma(W_{xf}x_t + W_{hf}h_{t-1} + W_{cf}c_{t-1} + b_f)$$

(4)

$$c_t = f_t c_{t-1} + i_t tanh(W_{xc}x_t + W_{hc}h_{t-1} + b_c)$$

(5)

$$o_t = \sigma(W_{xo}x_t + W_{ho}h_{t-1} + W_{co}c_t + b_o)$$

(6)

$$h_t = \sigma_t tanh(c_t)$$

(7)

Where σ is a sigmoid function, i, f, o, and c are input gate, forget gate, output gate, and cell activation vectors [28]. And they all have the same size as the hidden vector h.

In this study, we used BiLSTM which consists of two sub-networks of left and right sequence context that perform forward and backward pass respectively. We used element-by-element summation to combine forward and backward pass outputs. The output at position i is as follows:

$$h_i = \left[ \overrightarrow{h_i} \oplus \overleftarrow{h_i} \right]$$

(8)

(3) Kolmogorov-Arnold Networks (KANs) layer

Liu et al. proposed the Kolmogorov-Arnold Network (KAN) [21] to address the limitations of Multi-Layer Perceptrons (MLPs) in terms of parameter efficiency and interpretability. The intuition behind KAN lies in their utilization of the Kolmogorov-Arnold representation theorem, which positions learnable activation functions on the edges of the network (instead of the nodes) and eliminates traditional linear weights in favor of univariate functions (parametrized as splines) for each weight parameter. Similar to traditional MLPs, L-layer KAN can be seen as a nested structure of multiple KAN layers:

$$KAN(x) = (\Phi_{L-1} \circ \Phi_{L-2} \circ \cdots \circ \Phi_1 \circ \Phi_0)x$$

(13)

Where $\Phi_L$ is the function matrix corresponding to the Lth KAN layer, and $x0 \in R^{n0}$ is the input vector.

Specifically, for each KAN layer $\Phi_l$, a function matrix is defined, which consists of a set of learnable univariate functions. These functions can be represented as:

$$\Phi_l(x) = \{f_{l,1}(x), f_{l,2}(x), \cdots, f_{l,N_l}(x)\}$$

(14)

where $f_{l,i}$ is the univariate function of the i-th node in the l-th layer, and $N_l$ is the number of nodes in the l-th layer. Each $f_{l,i}$ can be further expressed as:

$$f_{l,i}(x) = \sum_{k=1}^{K} c_{l,i,k} B_k(x)$$

(15)

Here, $Bk(x)$ is the basis function, such as B-splines, and $C_{l,i,k}$ is the weight coefficient corresponding to the l-th layer, the i-th node, and the k-th basis function. These coefficients are learned and updated through backpropagation during the training process.

KAN offers a novel neural network architecture, whose assumption-free design can adapt to various input-output structures. Therefore, we integrate the KAN network into our model structure, merging the outputs of CNN and BiLSTM through residual connections to form the input of the KAN layer, and use the Adam optimizer to minimize the loss function.

(4) Advanced Techniques for Neural Network Optimization

From the model diagram Fig 1 presented earlier, it is evident that we employ residual connections, which enhance gradient propagation and optimize deep network performance [29]. Additionally, we utilize dropout to mitigate overfitting, and apply layer normalization [30] following the attention mechanism to enhance training stability and generalization capabilities.

(5) Parameter Optimization and Component Selection

The convolutional layers (ConvBlock1/2/3) were designed with kernel sizes optimized through extensive experimentation. ConvBlock1 uses kernels of sizes 9, 1, 5, and 8 to capture multi-scale features (e.g., large patterns with kernel 9, fine details with kernel 1).ConvBlock2/3 reduce kernel sizes (11–9, 1–8) to balance between computational efficiency and feature extraction capacity.Kernel size selection was guided by prior studies showing that smaller kernels (e.g., 3–5) are effective for short DNA sequences [25], while larger kernels can capture longer-range interactions. ReLU (Rectified Linear Unit) was chosen for all convolutional layers due to its efficiency in gradient propagation and computational speed. We systematically tested the number of hidden states of the BiLSTM module, examining various configurations including 8, 16, 32, 64, 128, and 256. By comparing the experimental results, we found that the BiLSTM module achieved the best performance when the number of hidden states was set to 128. In the design of the KAN layer, we also optimized the number of hidden layer nodes, examining different configurations such as 8, 16, 32, and 64. After comparative analysis, we finally decided to set the number of hidden layer nodes to 32, which showed the best performance in the experiment.

## Results

Our CBR-KAN model is implemented using PyTorch 2.5.0. The sample balance of 50 publicly available ChIP-seq data is 7:3 in the worst case and 1:1 in the best case. For each dataset, 10% of the entire data is selected as the validation set, 10% of the remaining 90% is used as the final test set, and the rest of the data is the training set. The data is stratified and sampled according to the label category ratio to ensure that the sample ratio of each category in the training set, validation set, and test set matches the complete dataset. The learning rate is 0.0005, the batch size is 64, the optimizer is Adam, and the epoch is set to 20. If the area under the receiver operating characteristic (ROC) curve (AUC) on the validation set does not improve after 5000 batches, the program will terminate the training early.

To ensure methodological rigor and computational efficiency, we adopted a two-stage experimental design: (1)Single-cellline validation: We first tested our model on GM12878, a well-characterized cell line with high-quality reference epigenomic data [30]. This allowed us to isolate the impact of our method from cross-cellline variability and technical noise. (2) Global benchmarking: After confirming performance in GM12878, we evaluated the model on a comprehensive dataset

encompassing multiple cell types. This two-step approach not only reduced redundant computations but also strengthened the generalizability claims of our work.

The model proposed in this paper is subjected to ablation experiments on the GM12878BATF dataset to compare the performance of each module and verify the effectiveness and necessity of adding these modules (i.e., CNN, BiLSTM, dan-2vec, and Residual). The experimental results are shown in Table 1.

For the experimental results in Table 1, we conduct the following analysis:

(1) Experimental comparison of convolution modules. Experiments were conducted on ConvBlock1, ConvBlock2, and ConvBlock3 of the convolution modules respectively. The experimental results show that the ConvBlock2 structure has the best effect. Then, the number of convolution kernel channels of ConvBlock2 was increased to verify whether the effect was improved due to the increase in the number of parameters. kernel*2 means increasing the number of convolution kernel channels to 2 times the original number, and kernel*3 means increasing the number of convolution

**Table 1. Performance of each module at GM12878BATF dataset.**

| Modules | model | ROC AUC | PR AUC | Precision | Recall | Accuracy | F1-score | MCC |
|---|---|---|---|---|---|---|---|---|
| ConvBlock | C-KAN (dna2vec) (ConvBlock3) | 0.919 | 0.839 | 0.803 | 0.720 | 0.872 | 0.759 | 0.674 |
| | C-KAN (dna2vec) (ConvBlock1) | 0.944 | 0.895 | 0.845 | 0.796 | 0.902 | 0.820 | 0.753 |
| | C-KAN (dna2vec) (ConvBlock2) | **0.950** | **0.896** | **0.884** | **0.770** | **0.907** | **0.823** | **0.763** |
| | C-KAN (dna2vec) (ConvBlock2)-kernel*2 | 0.950 | 0.901 | 0.826 | 0.842 | 0.905 | 0.834 | 0.768 |
| | C-KAN (dna2vec) (ConvBlock2)-kernel*3 | 0.949 | 0.889 | 0.843 | 0.833 | 0.910 | 0.838 | 0.776 |
| | C-KAN (dna2vec) C(ConvBlock1/2/3) | **0.953** | **0.910** | **0.887** | **0.815** | **0.919** | **0.849** | **0.795** |
| dna2vec | C-KAN (Non-dna2vec) C(ConvBlock1/2/3) | 0.947 | 0.897 | 0.799 | 0.856 | 0.899 | 0.826 | 0.756 |
| | C-KAN (dna2vec) C(ConvBlock1/2/3) | **0.953** | **0.910** | **0.887** | **0.815** | **0.919** | **0.849** | **0.795** |
| Residual MLP vs KAN | CB-KAN (dna2vec) C(ConvBlock1/2/3) (Non-Residual) | 0.936 | 0.881 | 0.732 | 0.860 | 0.872 | 0.791 | 0.704 |
| | CBR-MLP (dna2vec) C(ConvBlock1/2/3) (Residual) | 0.952 | 0.905 | 0.872 | 0.821 | 0.916 | 0.846 | 0.789 |
| | CBR-KAN (dna2vec) C(ConvBlock1/2/3) (Residual) | **0.954** | **0.910** | **0.860** | **0.851** | **0.919** | **0.855** | **0.799** |

kernel channels to 3 times the original number. The results show that simply increasing the number of convolution kernel channels cannot improve the model effect, while combining ConvBlock1, ConvBlock2, and ConvBlock3 (i.e., C(ConvBlock1/2/3)) will improve the performance of the model.

(2) Experimental comparison of dna2vec. Non-dna2vec means not using dna2vec for word embedding. The experimental results show that using dna2vec for word embedding will improve the model effect.

(3) Experimental comparisons were conducted on Residual and MLP. From the comparison between CB-KAN and CBR-KAN, it can be seen that the use of residual connections can greatly improve the model performance; from the comparison between CBR-MLP and CBR-KAN, it can be seen that the use of KAN is better than MLP.

1. Comparison with other Deep Learning Methods

Here, we compare the performance of the proposed CBR-KAN model with other traditional deep learning models. The deep learning models we used for performance analysis include: CNN, BiLSTM and BiGRU. The Precision, Recall, Accuracy, F1-score and MCC results are shown in Table 2, and the ROC AUC and PR AUC are shown in Fig 3. Of course, we also experimented with BiRNN, LSTM and GRU, but since these models cannot effectively learn patterns in the data to produce accurate predictions, the data is not shown here.

As can be seen from Table 2, when comparing the performance of CNN, Bi-LSTM, Bi-GRU and CBR-KAN models, the CBR-KAN model performs best in all evaluation indicators. The results on F1-score (0.855) and MCC (0.799) show that the model not only performs well in balancing precision and recall, but also provides a more accurate performance evaluation when all classification results are considered comprehensively.

As shown in Fig 3, the CBR-KAN model performs best in both ROC AUC (0.954) and PR AUC (0.910), which shows that the model performs well in distinguishing positive and negative samples and identifying positive samples.

2. Comparison with Existing Models

Here, we compare our proposed CBR-KAN model with existing methods. The Precision, Recall, Accuracy, F1-score and MCC results are shown in Table 3, and the ROC AUC and PR AUC are shown in Fig 4.

As can be seen from Table 3, the CBR-KAN model proposed in this paper performs well in multiple key performance indicators. Although its precision (Precision) 0.860 is not the highest, its accuracy (Accuracy) reaches 0.919, which is the highest among all models, indicating that CBR-KAN has extremely high accuracy in overall prediction. At the same time, the F1-score of CBR-KAN is 0.855, the highest among all models, indicating that the model has achieved a good balance between precision and recall. In addition, the Matthews correlation coefficient (MCC) of CBR-KAN is 0.799, which is also the highest among all models, which further confirms its advantage in classification performance. Therefore, although the accuracy is not the highest, the overall performance of the CBR-KAN model in the prediction task is very outstanding and has high reliability and effectiveness.

As shown in Fig 4, on the GM12878BATF dataset, CBR-KAN's ROC AUC reached 0.954, a high value that shows the model's excellent ability to identify positive and negative samples. In terms of the PR AUC indicator, CBR-KAN also leads

**Table 2. Comparison with deep learning algorithms at GM12878BATF dataset.**

| model | Precision | Recall | Accuracy | F1-score | MCC |
|---|---|---|---|---|---|
| CNN | 0.712 | 0.774 | 0.848 | 0.742 | 0.636 |
| Bi-LSTM | 0.787 | 0.618 | 0.845 | 0.692 | 0.598 |
| Bi-GRU | 0.807 | 0.684 | 0.865 | 0.741 | 0.655 |
| CBR-KAN | 0.860 | 0.851 | 0.919 | 0.855 | 0.799 |

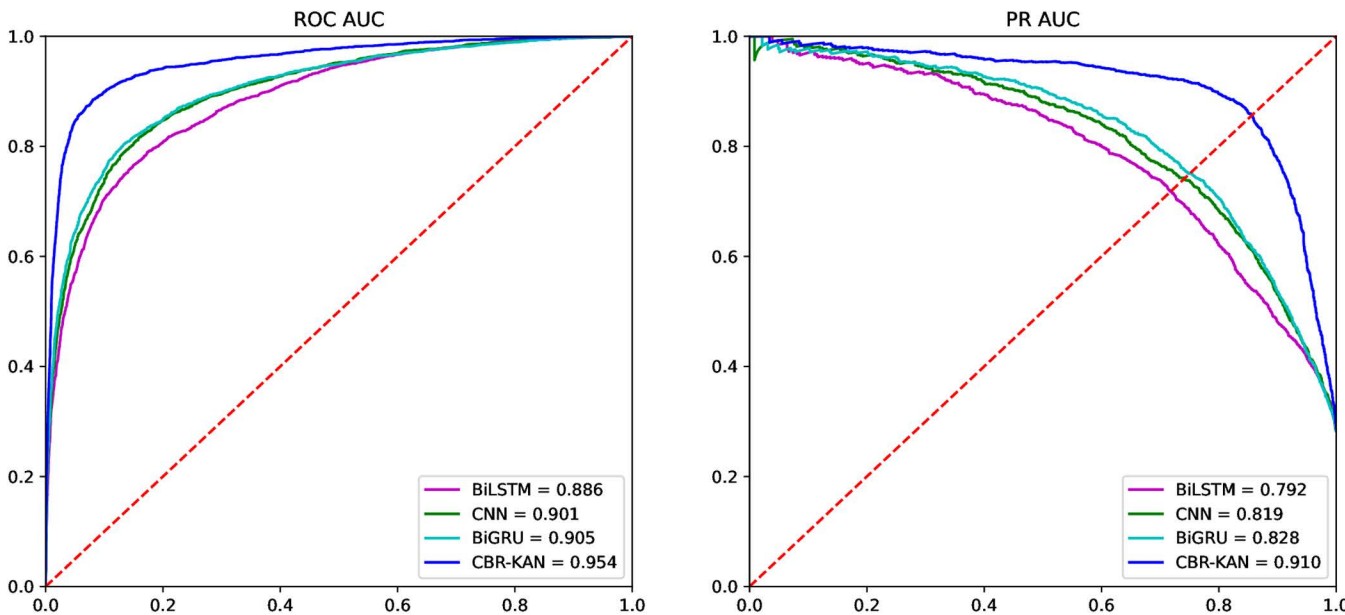

**Fig 3. Comparison of ROC AUC and PR AUC with deep learning algorithms.**

**Table 3. Comparison with existing models at GM12878BATF dataset.**

| model | Precision | Recall | Accuracy | F1-score | MCC |
|---|---|---|---|---|---|
| DeepBind | 0.810 | 0.639 | 0.856 | 0.714 | 0.627 |
| DanQ | 0.822 | 0.657 | 0.864 | 0.730 | 0.648 |
| DeepD2V | 0.866 | 0.690 | 0.883 | 0.768 | 0.699 |
| DeepSEA | 0.807 | 0.822 | 0.894 | 0.814 | 0.741 |
| CBR-KAN | 0.860 | 0.851 | 0.919 | 0.855 | 0.799 |

with a score of 0.910, which shows that it can better balance precision and recall and maintain high classification performance when dealing with unbalanced datasets. Compared with other models, DeepSEA's ROC AUC is 0.938 and PR AUC is 0.885, which is also quite good, but still slightly inferior to CBR-KAN. DeepD2V's ROC AUC is 0.927 and PR AUC is 0.860. Although it performs well in ROC AUC, it is slightly insufficient in PR AUC. DanQ's ROC AUC is 0.909 and PR AUC is 0.826, while DeepBind's ROC AUC is 0.897 and PR AUC is 0.810. Both models are relatively low in both indicators, indicating that there may be performance limitations in some aspects. Overall, the performance of the CBR-KAN model in both ROC AUC and PR AUC indicators exceeds that of other existing models, indicating the effectiveness of our proposed CBR-KAN model.

In order to comprehensively evaluate the effectiveness of the CBR-KAN model, we conducted experiments on 50 public ChIP-seq datasets (the GM12878BATF dataset used in the previous article is one of them) to ensure the stability and generalization ability of the model. The average results of the ROC AUC, Accuracy, F1-score and MCC indicators of each model on 50 data are shown in Fig 5, and the individual ROC AUC indicators of each model on each dataset are shown in Table 4. The scatter plot comparison results of each model and the CBR-KAN model in this paper on various datasets are shown in Fig 6.

As shown in Fig 5, on 50 public ChIP-seq datasets, the CBR-KAN model significantly outperforms DeepBind, DanQ, DeepD2V, and DeepSEA models in key performance indicators such as MCC, F1-score, accuracy, and ROC AUC.

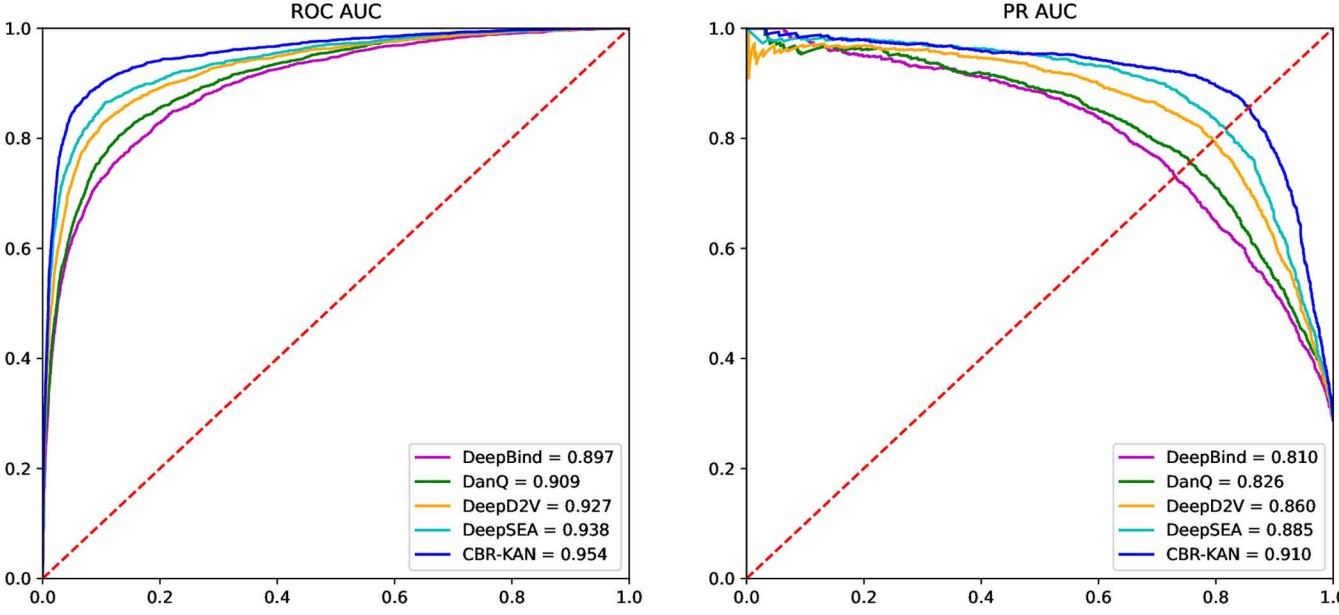

**Fig 4. Comparison of ROC AUC and PR AUC with existing models.**

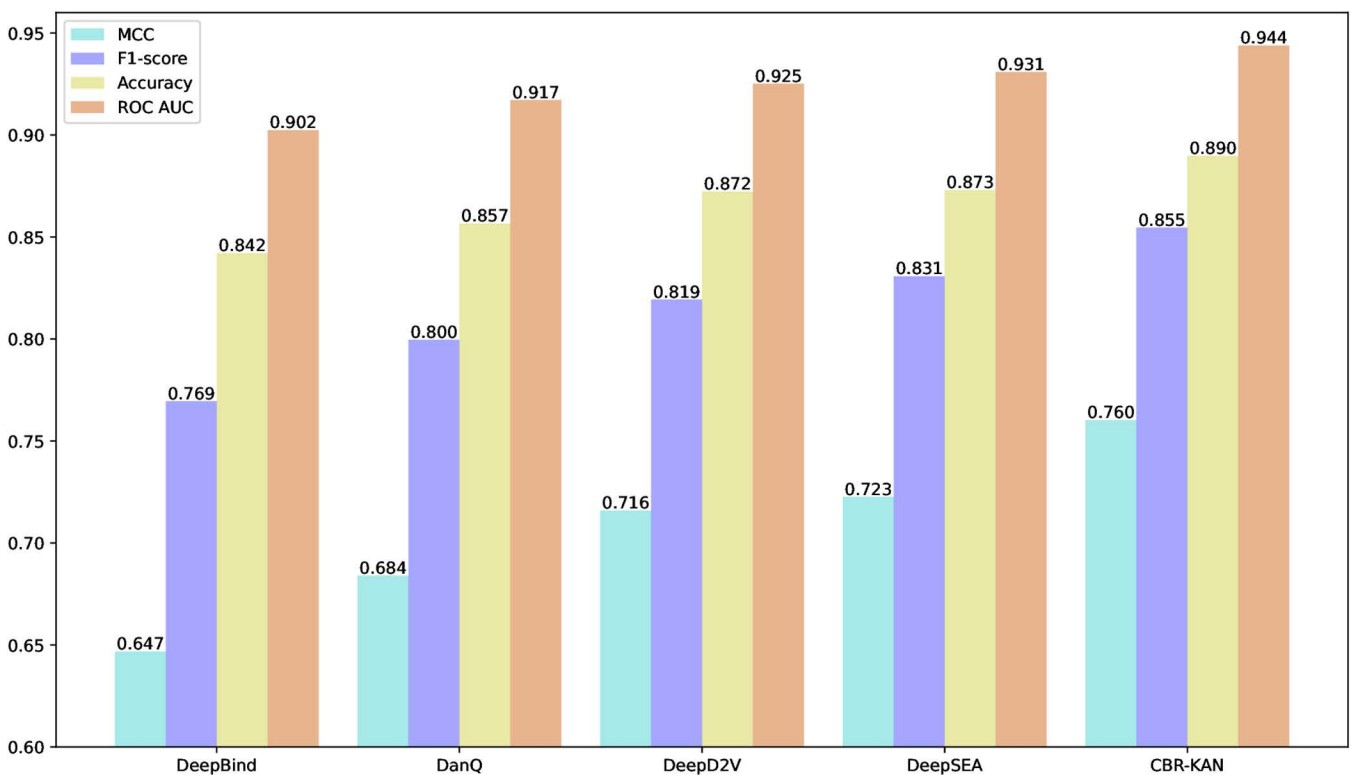

**Fig 5. Average results of each model on 50 datasets.**

**Table 4. ROC AUC of CBR-KAN with existing models for 50 datasets.**

| Cell Line | TF | DeepBind | DanQ | DeepD2V | DeepSEA | CBR-KAN |
|-----------|------|----------|-------|---------|---------|---------|
| Gm12878 | Batf | 0.897 | 0.909 | 0.927 | 0.938 | 0.954 |
| Gm12878 | Bcl11a | 0.853 | 0.857 | 0.500 | 0.884 | 0.918 |
| Gm12878 | Bcl3 | 0.880 | 0.901 | 0.930 | 0.904 | 0.935 |
| Gm12878 | Bclaf | 0.831 | 0.877 | 0.871 | 0.874 | 0.897 |
| Gm12878 | Ebf | 0.880 | 0.883 | 0.901 | 0.895 | 0.922 |
| Gm12878 | Egr1 | 0.939 | 0.959 | 0.963 | 0.965 | 0.968 |
| Gm12878 | Elf1 | 0.903 | 0.902 | 0.928 | 0.925 | 0.931 |
| Gm12878 | Ets1 | 0.909 | 0.940 | 0.957 | 0.947 | 0.959 |
| Gm12878 | Irf4 | 0.831 | 0.864 | 0.876 | 0.871 | 0.907 |
| Gm12878 | Mef2a | 0.812 | 0.898 | 0.922 | 0.911 | 0.931 |
| Gm12878 | Nrsf | 0.887 | 0.905 | 0.930 | 0.936 | 0.939 |
| Gm12878 | Pax5c20 | 0.859 | 0.876 | 0.888 | 0.892 | 0.907 |
| Gm12878 | Pax5n19 | 0.862 | 0.890 | 0.908 | 0.900 | 0.931 |
| Gm12878 | Pbx3 | 0.884 | 0.893 | 0.909 | 0.925 | 0.938 |
| Gm12878 | Pou2f2 | 0.847 | 0.869 | 0.869 | 0.878 | 0.908 |
| Gm12878 | Pu1 | 0.961 | 0.965 | 0.970 | 0.974 | 0.980 |
| Gm12878 | Rad21 | 0.949 | 0.973 | 0.987 | 0.987 | 0.987 |
| Gm12878 | Sp1 | 0.825 | 0.847 | 0.857 | 0.859 | 0.899 |
| Gm12878 | Srf | 0.871 | 0.902 | 0.945 | 0.931 | 0.948 |
| Gm12878 | Taf1 | 0.915 | 0.915 | 0.923 | 0.922 | 0.927 |
| Gm12878 | Tcf12 | 0.906 | 0.908 | 0.920 | 0.918 | 0.938 |
| Gm12878 | Usf1 | 0.937 | 0.951 | 0.957 | 0.959 | 0.960 |
| Gm12878 | Yy1 | 0.881 | 0.874 | 0.914 | 0.913 | 0.932 |
| H1hesc | Gabp | 0.895 | 0.899 | 0.942 | 0.927 | 0.943 |
| H1hesc | Nrsf | 0.903 | 0.938 | 0.962 | 0.960 | 0.967 |
| H1hesc | Rad21 | 0.963 | 0.964 | 0.978 | 0.982 | 0.982 |
| H1hesc | Sin3 | 0.917 | 0.923 | 0.934 | 0.934 | 0.938 |
| H1hesc | Sp1 | 0.893 | 0.898 | 0.921 | 0.910 | 0.935 |
| H1hesc | Srf | 0.937 | 0.916 | 0.969 | 0.963 | 0.975 |
| H1hesc | Taf1 | 0.926 | 0.931 | 0.932 | 0.933 | 0.936 |
| H1hesc | Tcf12 | 0.834 | 0.883 | 0.924 | 0.898 | 0.929 |
| H1hesc | Usf1 | 0.972 | 0.974 | 0.981 | 0.980 | 0.984 |
| H1hesc | Yy1 | 0.910 | 0.924 | 0.948 | 0.942 | 0.955 |
| K562 | Atf3 | 0.919 | 0.933 | 0.948 | 0.947 | 0.955 |
| K562 | E2f6 | 0.945 | 0.954 | 0.958 | 0.952 | 0.959 |
| K562 | Egr1 | 0.964 | 0.966 | 0.963 | 0.970 | 0.972 |
| K562 | Elf1 | 0.933 | 0.938 | 0.940 | 0.943 | 0.945 |
| K562 | Ets1 | 0.900 | 0.914 | 0.912 | 0.914 | 0.925 |
| K562 | Fosl1 | 0.945 | 0.964 | 0.965 | 0.962 | 0.968 |
| K562 | Gabp | 0.931 | 0.931 | 0.954 | 0.954 | 0.964 |
| K562 | Gata2 | 0.847 | 0.846 | 0.872 | 0.889 | 0.910 |
| K562 | Hey1 | 0.876 | 0.888 | 0.892 | 0.889 | 0.895 |
| K562 | Max | 0.929 | 0.936 | 0.942 | 0.938 | 0.947 |
| K562 | Nrsf | 0.876 | 0.918 | 0.930 | 0.922 | 0.942 |
| K562 | Pu1 | 0.979 | 0.983 | 0.985 | 0.985 | 0.988 |
| K562 | Rad21 | 0.968 | 0.976 | 0.985 | 0.991 | 0.989 |

*(Continued)*

**Table 4.** (Continued)

| Cell Line | TF | DeepBind | DanQ | DeepD2V | DeepSEA | CBR-KAN |
|-----------|------|----------|-------|---------|---------|---------|
| K562 | Srf | 0.857 | 0.907 | 0.932 | 0.930 | 0.940 |
| K562 | Taf1 | 0.920 | 0.926 | 0.937 | 0.934 | 0.938 |
| K562 | Usf1 | 0.947 | 0.953 | 0.964 | 0.962 | 0.966 |
| K562 | Yy1 | 0.913 | 0.920 | 0.936 | 0.926 | 0.938 |
| Avg | | 0.902 | 0.917 | 0.925 | 0.931 | 0.944 |

CBR-KAN has an MCC of 0.760, an F1-score of 0.855, an accuracy of 0.890, and a ROC AUC of 0.944. These results are due to its advanced network structure, including the combination of KAN network, CNN, BiLSTM, and residual connection, which enables it to more effectively capture complex features and long-distance dependencies in genomics tasks. In contrast, DeepBind scored relatively low in all indicators. Although it uses CNN and fully connected layers, the simplicity of its structure limits its ability to capture complex features. DanQ enhances the processing of bidirectional sequence information by introducing BiLSTM, but its single-layer CNN still has limitations in processing complex features. DeepD2V improves feature extraction capabilities by increasing the number of CNN layers and combines BiLSTM to process sequence information, but the complexity of the model may lead to training challenges. DeepSEA relies entirely on multi-layer CNNs. Although it can extract hierarchical features, it is insufficient in processing sequence information. In summary, the CBR-KAN model proposed in this paper has shown excellent performance in multiple performance indicators, proving its effectiveness and generalization ability.

As shown in Fig 6, we use a scatter plot to show the comparison results of the models on various data sets. The point on the upper left corner of the diagonal line in the figure indicates that the CBR-KAN model is better than the comparison method. As can be seen from the figure, in the test of 50 public ChIP-seq data sets, the CBR-KAN model is significantly better than DeepBind and DanQ in terms of ROC AUC, PR AUC and F1 score performance indicators, showing excellent balance and sorting capabilities. Although CBR-KAN is slightly inferior to DeepD2V and DeepSEA in certain specific data sets, overall CBR-KAN is still better than DeepD2V and DeepSEA in these indicators, showing higher consistency and reliability. Overall, CBR-KAN has achieved good performance in various performance indicators.

As shown in Table 4, the CBR-KAN model performed the best among all the models involved in the comparison, with an average ROC AUC value of 0.944, significantly higher than DeepBind, DanQ, DeepD2V and DeepSEA models. This result highlights the efficiency and stability of CBR-KAN in genomic classification tasks. Although DeepSEA also showed strong performance on multiple datasets, the comprehensive advantages of CBR-KAN, especially on datasets such as Batf and Pu1 of Gm12878, proved its superiority in processing various types of genomic data. Other models, such as DeepBind and DanQ, performed relatively weakly on some datasets, which may be related to their limitations in feature extraction and model structure. DeepD2V showed good generalization ability, but its overall performance was still inferior to CBR-KAN. Therefore, the classification ability of the CBR-KAN model on genomic datasets has been fully verified, showing its potential and practicality as an advanced model.

Furthermore, we conducted a T-test on the accuracy results under different methods. The results showed that the p values between the results of CBR-KAN and those of "DeepBind", "DanQ", "DeepD2V", and "DeepSEA" were $1.012 \times 10^{-7}$, $5.4054 \times 10^{-5}$, 0.0422, and 0.0398 respectively, all of which were significant.

To further validate the robustness and generalisability of our model across diverse cell lines, we sourced data from three additional cell lines available on GEO: GSM1873040 (LNCaP), GSM2437758 (MCF - 7), and GSM2067524 (A549). Subsequently, we applied our model to these datasets, and the outcomes are presented in Table 5.

As shown in Table 5, the CBR-KAN model proposed in this study demonstrates significant superiority over existing models across all core metrics (ROC AUC, accuracy, F1-score, MCC) on three cell line datasets: LNCaP, A549, and

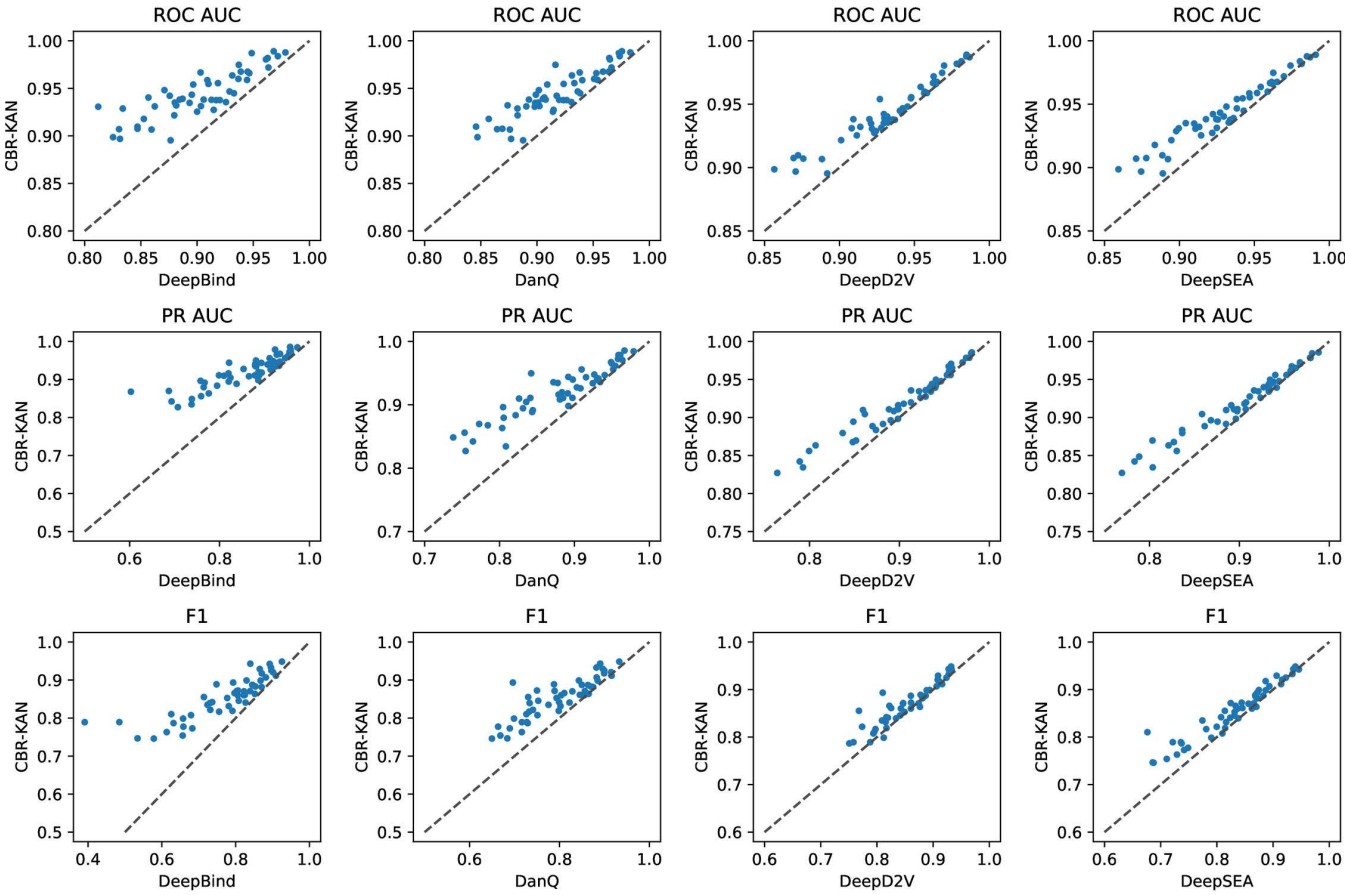

**Fig 6. Comparison of AUC and F1-score indicators with existing models on 50 datasets.**

**Table 5. Comparison with existing models across diverse cell lines.**

| Datasets | Model | ROC AUC | Accuracy | F1-score | MCC |
|---|---|---|---|---|---|
| GSM1873040 (LNCaP) | DeepBind | 0.935 | 0.915 | 0.735 | 0.693 |
| | DanQ | 0.948 | 0.923 | 0.753 | 0.722 |
| | DeepD2V | 0.961 | 0.936 | 0.805 | 0.772 |
| | DeepSEA | 0.957 | 0.934 | 0.813 | 0.773 |
| | **CBR-KAN** | **0.964** | **0.941** | **0.821** | **0.792** |
| GSM2437758 (A549) | DeepBind | 0.968 | 0.936 | 0.851 | 0.812 |
| | DanQ | 0.981 | 0.961 | 0.911 | 0.887 |
| | DeepD2V | 0.990 | 0.977 | 0.947 | 0.932 |
| | DeepSEA | 0.991 | 0.976 | 0.945 | 0.930 |
| | **CBR-KAN** | **0.993** | **0.977** | **0.948** | **0.934** |
| GSM2067524 (MCF-7) | DeepBind | 0.976 | 0.928 | 0.879 | 0.829 |
| | DanQ | 0.981 | 0.936 | 0.882 | 0.840 |
| | DeepD2V | 0.992 | 0.961 | 0.927 | 0.902 |
| | DeepSEA | 0.991 | 0.963 | 0.934 | 0.908 |
| | **CBR-KAN** | **0.993** | **0.968** | **0.944** | **0.922** |

MCF-7, validating its exceptional robustness and cross-cell-line generalizability. CBR-KAN achieves the highest ROC AUC values (LNCaP: 0.964 vs. 0.961; A549: 0.993 vs. 0.991; MCF-7: 0.993 vs. 0.992) and MCC scores (e.g., 0.922 vs. 0.902 in MCF-7), demonstrating its enhanced capability to capture complex genomic sequence features. The model exhibits particularly notable advantages in class-imbalanced scenarios, as evidenced by a 2.4% relative improvement in MCC compared to the second-best model on the LNCaP cell line dataset.

## Discussion

Our research results indicate that the CBR-KAN model significantly outperforms other state-of-the-art methods in predicting transcription factor binding sites across multiple ChIP-seq datasets. The performance of CBR-KAN is markedly superior to DeepBind, DanQ, DeepD2V, and DeepSEA, as reflected in metrics such as ROC AUC, PR AUC, accuracy, and F1 score. For instance, in the GM12878 cell line, the ROC AUC for the BATF transcription factor reached 0.954 for CBR-KAN, compared to 0.897 for DeepBind, 0.909 for DanQ, 0.927 for DeepD2V, and 0.938 for DeepSEA. This significant improvement can be attributed to several key architectural innovations of CBR-KAN. Firstly, the integration of the KAN network, which replaces the traditional multilayer perceptron (MLP), enhances the model's robustness and feature extraction capabilities [31]. The nested structure of the KAN network allows for more efficient parameter utilization and increased interpretability, which is crucial for dealing with the complexity and noise of ChIP-seq data. Secondly, the combination of convolutional neural networks (CNN) and bidirectional long short-term memory (BiLSTM) networks enables CBR-KAN to capture spatial and temporal features in DNA sequences. The CNN layers extract hierarchical features from DNA sequences, while the BiLSTM layers capture long-term dependencies and sequence information. This dual approach ensures that the model can effectively identify transcription factor binding motifs even in the presence of noise. Lastly, the introduction of residual connections addresses the vanishing gradient problem that often plagues deep neural networks. By learning residuals between network layers, residual connections facilitate the optimization process and stabilize training, leading to more robust and accurate predictions.

The performance of CBR-KAN varies across different datasets, reflecting the unique characteristics of each cell line and transcription factor. For example, in the H1-hESC cell line, the ROC AUC for the NRSF transcription factor reached 0.967 for CBR-KAN, compared to 0.960 for DeepSEA. The higher performance in H1-hESC indicates that CBR-KAN is particularly effective on datasets with high-quality ChIP-seq data and well-defined transcription factor features. Conversely, in the K562 cell line, the ROC AUC for the NRSF transcription factor was 0.942 for CBR-KAN and 0.922 for DeepSEA. The slightly lower performance in K562 suggests that CBR-KAN may face challenges on datasets with higher noise or less distinct transcription factor features. However, even on these challenging datasets, CBR-KAN still outperforms other methods, demonstrating its robustness and generalization capabilities. The results of this study not only validate the effectiveness of CBR-KAN in predicting transcription factor binding sites but also highlight its broader impact in bioinformatics. The integration of advanced neural network architectures, such as KAN networks, and residual connections, provides a promising direction for developing more effective models for various genomic problems. For instance, in the analysis of single cell sequencing data, predicting transcription factor binding sites at the single cell level contributes to a more detailed understanding of gene regulation in complex tissues.

In future research, the potential applications of transfer learning, Transformer models, and large-scale pre-trained models in predicting transcription factor binding sites may further enhance the accuracy and efficiency of these models. Transfer learning has been widely applied and has achieved good results in various fields in recent years. For example, models such as BERT and ERNIE have emerged in the field of natural language processing, and models such as VGG and ResNet have emerged in the field of computer vision. Similarly, in the near future, transfer learning may also be used to predict DNA transcription factor binding sites [32]. Models can be trained on related datasets and fine-tuned by transferring knowledge or loading pre-trained models. Feature transfer not only enhances the model's generalization ability but also greatly reduces training time. Even when applied to tasks with low similarity, transfer learning can produce

 

better results than randomly initializing parameters. In addition, Transformer models, which have achieved great success in natural language processing and image processing, are expected to improve the prediction accuracy of transcription factor binding sites [33]. The self-attention mechanism of Transformers can effectively capture long-range dependencies in sequences, which is crucial for processing large-scale DNA sequences. With the continuous advancement of computing power and data scale, the application prospects of large language models in bioinformatics are very broad [34]. These models show good generalization ability and are adept at capturing complex patterns, which is crucial for predicting complex protein-DNA interactions in genomics [35].

Although the CBR-KAN model has shown significant progress in predicting transcription factor binding sites (TFBS), it is crucial to acknowledge and discuss its limitations and the challenges that still exist in this field. A long-standing challenge in deep learning, including the CBR-KAN model, is the issue of interpretability and explainability. Deep neural networks with complex architectures like CBR-KAN operate like "black boxes", making it difficult to understand the specific features or patterns the model is learning. In fields such as bioinformatics, this lack of interpretability can be a problem because understanding the biological significance of model predictions is crucial. Although the CBR-KAN model performed well on the ChIP-seq datasets used in this study, its generalization ability to new and unseen data remains a key challenge. The model's performance on datasets from different cell types, conditions, or species has not been widely evaluated. Ensuring that the model can generalize well to different genomic backgrounds is essential for its practical application in real-world scenarios.

## Conclusion

This study presents a KAN-based hybrid deep neural network, CBR-KAN, which demonstrates significant advantages in the accurate identification of transcription factor binding sites. CBR-KAN takes the segmented embedding vectors of DNA sequences as input and captures local features and long-term dependencies in DNA sequences by integrating CNN and BiLSTM. At the key output layer of the model, CBR-KAN employs an advanced KAN network structure, innovatively replacing the traditional MLP network, significantly improving the model's accuracy in predicting transcription factor binding sites. Additionally, we introduce residual connections to enhance the model's robustness and stability. In comparative experiments across 50 ChIP-seq datasets, the CBR-KAN model excels in multiple key performance indicators, significantly outperforming other deep learning methods.

## Author contributions

**Data curation:** Huijun Hao.

**Methodology:** Guodong He.

**Project administration:** Wei Chen.

**Supervision:** Wei Chen.

**Validation:** Guodong He, Jiahao Ye.

**Visualization:** Guodong He, Jiahao Ye.

**Writing – original draft:** Guodong He.

**Writing – review & editing:** Wei Chen.

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
