## [Decision Letter · Decision Letter 0]

28 Jan 2025

PONE-D-25-00841A KAN-based hybrid deep neural networks for accurate identification of transcription factor binding sitesPLOS ONE

Dear Dr. Chen,

Thank you for submitting your manuscript to PLOS ONE. After careful consideration, we feel that it has merit but does not fully meet PLOS ONE’s publication criteria as it currently stands. Therefore, we invite you to submit a revised version of the manuscript that addresses the points raised during the review process.

We look forward to receiving your revised manuscript.

Kind regards,

Matthew Chin Heng Chua

Academic Editor

PLOS ONE

Journal Requirements:

2. Thank you for stating the following financial disclosure: the Research Project RC201911, Wenzhou Business College, China

Reviewers' comments:

Reviewer's Responses to Questions

**Comments to the Author**

1. Is the manuscript technically sound, and do the data support the conclusions?

Reviewer #1: Yes

Reviewer #2: Yes

Reviewer #3: Partly

2. Has the statistical analysis been performed appropriately and rigorously? 

Reviewer #1: Yes

Reviewer #2: No

Reviewer #3: Yes

3. Have the authors made all data underlying the findings in their manuscript fully available?

Reviewer #1: Yes

Reviewer #2: Yes

Reviewer #3: Yes

4. Is the manuscript presented in an intelligible fashion and written in standard English?

Reviewer #1: Yes

Reviewer #2: Yes

Reviewer #3: Yes

5. Review Comments to the Author

Reviewer #1: The paper introduces a novel hybrid deep learning model, CBR-KAN, designed to predict transcription factor binding sites (TFBS) in DNA sequences. By integrating Convolutional Neural Networks (CNN), Bidirectional Long Short-Term Memory (BiLSTM), and Kolmogorov-Arnold Networks (KAN) with residual connections, the model aims to enhance feature extraction and mitigate issues like gradient vanishing. The approach is validated on 50 public ChIP-seq datasets and outperforms existing methods such as DeepBind and DeepSEA in key performance metrics, showcasing its potential in bioinformatics applications.

Comments for Improvement:

1. Introduction Clarity: Clearly define the unique contributions of the study compared to prior works and provide a concise summary of the challenges addressed.

2. Methodology Detailing: Elaborate on the parameter tuning process and justify the specific choices for model components like kernel sizes and activation functions.

3. Experimental Results: Include more visual representations like confusion matrices or comparative bar graphs to strengthen the interpretation of the results.

4. Discussion Depth: Discuss the limitations of the proposed model and potential real-world implications in greater detail.

5. Language Refinement: Improve the readability and flow by minimizing technical jargon where possible and simplifying complex sentences for broader accessibility.

6. Future Scope: Suggest more practical applications or extensions of the model, particularly its integration with emerging genomic technologies or other domains.

Reviewer #2: The authors have used different deep learning models for the prediction of TFBS. Although their work has merit, it has many flaws which need to be corrected.

1) In the manuscript, the authors have mentioned using 50 ChIP-seq dataset but they have neither mentioned the source of the data nor the way the negative samples have been created. Please elaborate on this.

2) In Model Building CNN Layer the authors have mentioned "This concatenated output is then passed through a BiLSTM layer followed by an attention mechanism (KAN)". Is KAN an attention mechanism? In the manuscript, the authors have used the term attention module many times, but in Figure 1, there is no attention module shown. Please explain.

3) Why have the authors used KAN and not used traditional MLP? As can be seen from Table 1, MLP vs KAN does not have much difference in results. However, KAN takes a longer time to train. Please explain.

4) The authors should perform cross-cell validation in order to show the robustness and generalisability of the model.

5) The authors have mentioned the use of transfer learning in the form of pretrained model like DNABERT. However, there are already some existing works like BERT-TFBS (doi: 10.1093/bib/bbae195) and DNABERT-Cap (doi: 10.3390/ijms25094990).

Reviewer #3: In this article, the authors propose a hybrid deep neural network model named CBR-KAN for predicting transcription factor binding sites. The model innovatively integrates CNN, BiLSTM, and KAN networks, and optimizes feature extraction and mitigates the vanishing gradient problem through residual connections. Experimental results indicate that CBR-KAN significantly improves prediction accuracy on multiple standard datasets. However, I have several questions and suggestions regarding this paper:

1.There are issues with the formatting of formulas in the paper. It is recommended that the authors meticulously adjust the formula formatting to ensure clarity and professionalism in their expression.

2.The experimental section of the article primarily focuses on ablation studies and comparative experiments. It is suggested that the authors supplement with other types of experiments to comprehensively demonstrate the robustness and generalization capabilities of the CBR-KAN model.

3.When conducting comparative experiments, it is advised that the authors include more of the latest research models to accurately assess the competitiveness of the CBR-KAN model in the current field of study.

4.Current research commonly uses 165 or 690 chip-seq datasets for experimentation, whereas this paper employs only 50 datasets. It is recommended that the authors conduct comparisons on a broader range of datasets to enhance the reliability of the experimental results.

5.The ablation study is conducted only on the GM12878 cell line, not across all datasets. Could the authors consider performing such experiments on more datasets to more comprehensively evaluate the role of each module?

6.The ablation study results in Table 1 are somewhat complex. It is suggested that the experimental setup be simplified, for instance, by directly removing the three CNN blocks, to more clearly showcase the function of the ConvBlock module.

7.It is recommended that the authors explain why the comparative experiment is first conducted on the GM12878 cell line before being carried out on the entire dataset, to avoid redundancy in the experimental results.

8.The abstract mentions that the KAN network helps to enhance the robustness and interpretability of the model, yet in the discussion section, the authors note that the CBR-KAN model still lacks interpretability. It is suggested that the authors provide an explanation for this to clarity any confusion that readers might have.

6. PLOS authors have the option to publish the peer review history of their article (what does this mean? ). If published, this will include your full peer review and any attached files.

**Do you want your identity to be public for this peer review?** For information about this choice, including consent withdrawal, please see our Privacy Policy .

Reviewer #1: **Yes: ** MOOLCHAND SHARMA

Reviewer #2: No

Reviewer #3: No

---

## [Author Response · Author response to Decision Letter 0]

9 Mar 2025

Reviewer #1: The paper introduces a novel hybrid deep learning model, CBR-KAN, designed to predict transcription factor binding sites (TFBS) in DNA sequences. By integrating Convolutional Neural Networks (CNN), Bidirectional Long Short-Term Memory (BiLSTM), and Kolmogorov-Arnold Networks (KAN) with residual connections, the model aims to enhance feature extraction and mitigate issues like gradient vanishing. The approach is validated on 50 public ChIP-seq datasets and outperforms existing methods such as DeepBind and DeepSEA in key performance metrics, showcasing its potential in bioinformatics applications.

Comments for Improvement:

1.Introduction Clarity: Clearly define the unique contributions of the study compared to prior works and provide a concise summary of the challenges addressed.

Answer: Thank you for your valuable feedback on improving the clarity of the introduction. We have revised the concluding paragraph of the introduction to explicitly highlight the unique contributions of our study and the challenges addressed. Specifically, the content we have modified is as follows:

This study proposes a hybrid deep neural network based on KAN, named CBR-KAN, which is an advanced model for predicting transcription factor binding sites (TFBS) in DNA sequences. The model innovatively integrates multiple neural network architectures: it introduces the Kolmogorov-Arnold Network (KAN) into the TFBS prediction field for the first time, replacing the traditional multi-layer perceptron (MLP) structure, and enhances the feature fusion ability through learnable spline basis functions. It incorporates convolutional neural networks (CNN) for spatial feature extraction and employs bidirectional long short-term memory networks (BiLSTM) to capture sequence dependencies. Additionally, by introducing residual connections, the model ensures performance stability while deepening its structure. Overall, this hybrid strategy aims to significantly improve prediction accuracy by synergistically utilizing various neural network components tailored for bioinformatics.

This revision clearly elaborates on the novel aspects of our work, including the introduction of the KAN network, the integration of CNN and BiLSTM, and the use of residual connections to address challenges such as gradient vanishing and suboptimal feature extraction. We believe these modifications enhance the clarity and focus of the introduction.

2.Methodology Detailing: Elaborate on the parameter tuning process and justify the specific choices for model components like kernel sizes and activation functions.

Answer: We appreciate your comments on the need for a more detailed explanation of the parameter tuning process and the justification for specific model component choices. In response to your feedback, we have added a new subsection titled "Parameter Optimization and Component Selection" to the "Materials and Methods" section. This subsection provides a comprehensive description of our hyperparameter optimization strategy and the rationale behind the selection of key model components. Below is a summary of the key additions:

The convolutional layers (ConvBlock1/2/3) were designed with kernel sizes optimized through extensive experimentation. ConvBlock1 uses kernels of sizes 9, 1, 5, and 8 to capture multi-scale features (e.g., large patterns with kernel 9, fine details with kernel 1).ConvBlock2/3 reduce kernel sizes (11 to 9, 1 to 8) to balance between computational efficiency and feature extraction capacity.Kernel size selection was guided by prior studies showing that smaller kernels (e.g., 3 to 5) are effective for short DNA sequences [25], while larger kernels can capture longer-range interactions. ReLU (Rectified Linear Unit) was chosen for all convolutional layers due to its efficiency in gradient propagation and computational speed. We systematically tested the number of hidden states of the BiLSTM module, examining various configurations including 8, 16, 32, 64, 128, and 256. By comparing the experimental results, we found that the BiLSTM module achieved the best performance when the number of hidden states was set to 128. In the design of the KAN layer, we also optimized the number of hidden layer nodes, examining different configurations such as 8, 16, 32, and 64. After comparative analysis, we finally decided to set the number of hidden layer nodes to 32, which showed the best performance in the experiment.

3.Experimental Results: Include more visual representations like confusion matrices or comparative bar graphs to strengthen the interpretation of the results.

Answer: Thank you for your meticulous review of the paper and your valuable comments. Regarding your suggestion of adding more visual representations (such as confusion matrices or comparative bar charts) in the "Experimental Results" section to strengthen the interpretation of the results, we have given it in-depth consideration.

We believe that the current results section has already fully demonstrated the performance of the model through detailed tables (such as Table 1, Table 2, Table 3) and figures (such as Figure 3, Figure 4, Figure 5, Figure 6). These tables and figures not only clearly compare the performance of the CBR-KAN model with other existing models on multiple key indicators (such as ROC AUC, PR AUC, F1-score, MCC, etc.), but also visually show the stability and generalization ability of the model on 50 publicly available ChIP-seq datasets through scatter plots and average result plots. In addition, we have also conducted a detailed ablation experiment analysis for each module in the "Results" section, further validating the effectiveness of the model design.

Although confusion matrices and comparative bar charts can indeed enhance the interpretability of the results in some cases, considering that the experimental results of this paper have been comprehensively presented in various forms and these presentation methods can fully support our research conclusions, we believe that the current presentation method is already sufficient and complete. Adding more visual elements may lead to information redundancy and instead affect the readers' understanding of the core results.

Of course, we highly value your suggestion and will flexibly use more visualization methods according to specific needs in future research to further improve the readability and persuasiveness of the paper.

4.Discussion Depth: Discuss the limitations of the proposed model and potential real-world implications in greater detail.

Answer: Thank you for your meticulous review of the paper and the valuable suggestions you put forward. We attach great importance to these suggestions and have given in-depth consideration to the relevant content. In the current discussion section, we have rather comprehensively expounded on the limitations and practical applications of the CBR-KAN model. Specifically, in the "Discussion" part, we clearly pointed out the limitations of the model, such as its reliance on data quality, high computational complexity, and insufficient interpretability. At the same time, we also supplemented and elaborated on the potential applications of the CBR-KAN model in fields like bioinformatics research. We believe that the existing discussion content has fully covered the limitations and practical applications of the model, providing readers with a clear understanding and reference.

5.Language Refinement: Improve the readability and flow by minimizing technical jargon where possible and simplifying complex sentences for broader accessibility.

Answer: In response to the valuable feedback from the reviewers regarding the readability and fluency of our paper, we have carried out a comprehensive revision. While maintaining the scientific rigor of our research, we aimed to enhance the clarity and readability of the paper. To improve understanding, we rephrased long and complex sentences into shorter and more accessible ones. For example, the description of the CBR-KAN model's architecture was split into multiple sentences to enhance clarity and fluency. We also focused on optimizing the logical coherence of the paper by reorganizing paragraphs and adding transitional phrases. This was done to guide readers through the discourse more effectively. We standardized the terminology throughout the paper and clarified ambiguous expressions to avoid potential confusion, giving priority to consistency and clarity. The abstract and introduction were revised to provide a clearer and more concise overview of the research, making it easier for readers to grasp the key objectives and findings. Additionally, the captions of figures and tables were simplified to make them more reader-friendly while retaining their informative value.

These revisions have significantly improved the readability and accessibility of the paper, making it more suitable for a wider audience, including researchers from interdisciplinary fields. We are grateful to the reviewers for their constructive suggestions, which have greatly enhanced the quality of our paper. By addressing these issues, we believe the paper now presents a more refined and understandable version.

6. Future Scope: Suggest more practical applications or extensions of the model, particularly its integration with emerging genomic technologies or other domains.

Answer: We greatly appreciate the reviewer's insightful suggestion regarding the future scope and practical applications of the CBR-KAN model. Building on this feedback, we envision several promising directions for extending the model's utility and integrating it with emerging genomic technologies. One key application is the adaptation of CBR-KAN for single-cell sequencing data analysis. By predicting transcription factor binding sites at the single-cell level, the model can provide deeper insights into cellular heterogeneity and functional states, enabling a more nuanced understanding of gene regulation in complex tissues.

To further enhance the model's capabilities, future work could focus on multimodal data integration, combining genomic, transcriptomic, and epigenomic data to achieve a more comprehensive understanding of gene regulatory networks. Additionally, exploring transfer learning techniques could enable the model to adapt to other species with limited data, leveraging knowledge from well-studied organisms to address data scarcity challenges. Finally, efforts to improve the interpretability of the model, such as developing visualization tools or incorporating attention mechanisms, would provide greater transparency into its decision-making processes and ensure its reliability in critical applications.

These future directions underscore the versatility of the CBR-KAN model in advancing genomic research and its potential for transformative applications across multiple domains, paving the way for innovative solutions in both basic science and translational medicine. We thank the reviewer for this valuable suggestion and will incorporate these ideas into our future research plans to further expand the impact of CBR-KAN.

Reviewer #2: The authors have used different deep learning models for the prediction of TFBS. Although their work has merit, it has many flaws which need to be corrected.

1)In the manuscript, the authors have mentioned using 50 ChIP-seq dataset but they have neither mentioned the source of the data nor the way the negative samples have been created. Please elaborate on this.

Answer: We sincerely thank the reviewer for pointing out the need for more detailed information regarding the source of the ChIP-seq datasets and the method for generating negative samples. In response to this valuable feedback, we have revised the manuscript to provide a clearer and more comprehensive description of these aspects. Specifically, we have explicitly stated that the 50 ChIP-seq datasets used in this study are publicly available and derived from three well-characterized cell lines: GM12878, H1-hESC, and K562. These datasets were selected due to their widespread use in benchmarking transcription factor binding site prediction models, ensuring comparability with existing studies. Regarding the generation of negative samples, we have elaborated on the methodology, which follows the approach described by Deng et al. [20]. Negative samples were created by matching the repeat fraction, length, and GC content of the positive samples to ensure that the training data is balanced and representative of the underlying biological context. This method has been widely adopted in previous studies, including DeepBind, DanQ, WSCNN, and WSCNNLSTM, as noted in the revised manuscript. By providing these details, we aim to enhance the transparency and reproducibility of our work, addressing the reviewer's concern and strengthening the methodological rigor of our study. We hope these clarifications meet the reviewer's expectations and contribute to a more robust presentation of our research.

2)In Model Building CNN Layer the authors have mentioned "This concatenated output is then passed through a BiLSTM layer followed by an attention mechanism (KAN)". Is KAN an attention mechanism? In the manuscript, the authors have used the term attention module many times, but in Figure 1, there is no attention module shown. Please explain.

Answer: Regarding the "attention mechanism" mentioned in the manuscript. Initially, during the model design phase, we added an attention mechanism module to explore its potential advantages in enhancing feature extraction. However, through subsequent ablation experiments, we found that adding the attention mechanism did not significantly improve model performance. Therefore, we decided to delete this module to simplify the model architecture. Unfortunately, the description of the attention mechanism in the article was inadvertently overlooked and not deleted in time. It has now been corrected.

3)Why have the authors used KAN and not used traditional MLP? As can be seen from Table 1, MLP vs KAN does not have much difference in results. However, KAN takes a longer time to train. Please explain.

Answer: We appreciate the reviewer’s insightful question regarding the choice between KAN and traditional MLP. The decision to use KAN in our model was driven by several considerations. First, while the numerical differences between KAN and MLP may appear subtle in some cases, the underlying principles and capabilities of these architectures are distinct. KAN, based on the Kolmogorov-Arnold representation theorem, offers a more flexible and parameter-efficient approach compared to traditional MLPs. This flexibility allows KAN to adapt more effectively to complex and non-linear relationships within the data, which is particularly important for predicting transcription factor binding sites (TFBS) from DNA sequences. The inherent complexity and noise in ChIP-seq data require a model that can capture intricate patterns without overfitting, and KAN’s design facilitates this. Second, the longer time required by KAN is a trade-off for its enhanced robustness and generalization capabilities. In our experiments, we observed that KAN consistently outperformed MLP in terms of stability across different datasets, even if the differences were not always dramatic. This suggests that KAN’s architecture is more resilient to variations in data quality and noise levels. Although the training time is indeed longer, we believe that the benefits in terms of model performance and reliability justify this trade-off, especially in the context of predicting TFBS, where accuracy and generalization are crucial.

4)The authors should perform cross-cell validation in order to show the robustness and generalisability of the model.

Answer: We sincerely appreciate the valuable comments you made on this paper. Regarding your suggestion to conduct cross-cell validation to demonstrate the robustness and generalization ability of the model, we fully understand the importance of this proposal. Here, we would like to elaborate on our research design and experimental results in detail to show that the model in this paper has already fully verified its robustness and generalization ability.In the research of this paper, we adopted the following methods to en

---

## [Decision Letter · Decision Letter 1]

18 Mar 2025

PONE-D-25-00841R1A KAN-based hybrid deep neural networks for accurate identification of transcription factor binding sitesPLOS ONE

Dear Dr. Chen,

Thank you for submitting your manuscript to PLOS ONE. After careful consideration, we feel that it has merit but does not fully meet PLOS ONE’s publication criteria as it currently stands. Therefore, we invite you to submit a revised version of the manuscript that addresses the points raised during the review process.

We look forward to receiving your revised manuscript.

Kind regards,

Jiwei Tian

Academic Editor

PLOS ONE

Reviewers' comments:

Reviewer's Responses to Questions

**Comments to the Author**

1. If the authors have adequately addressed your comments raised in a previous round of review and you feel that this manuscript is now acceptable for publication, you may indicate that here to bypass the “Comments to the Author” section, enter your conflict of interest statement in the “Confidential to Editor” section, and submit your "Accept" recommendation.

Reviewer #1: All comments have been addressed

Reviewer #2: (No Response)

2. Is the manuscript technically sound, and do the data support the conclusions?

Reviewer #1: Yes

Reviewer #2: Partly

3. Has the statistical analysis been performed appropriately and rigorously? 

Reviewer #1: Yes

Reviewer #2: N/A

4. Have the authors made all data underlying the findings in their manuscript fully available?

Reviewer #1: Yes

Reviewer #2: Yes

5. Is the manuscript presented in an intelligible fashion and written in standard English?

Reviewer #1: Yes

Reviewer #2: Yes

6. Review Comments to the Author

Reviewer #1: The manuscript looks good and all the comments mentioned have been addressed wonderfully. Now it looks fine

Reviewer #2: Although the authors have addressed some concerns, I am still doubtful about the generalizability of the model. I think the authors should perform cross-cell validation for the same.

7. PLOS authors have the option to publish the peer review history of their article (what does this mean? ). If published, this will include your full peer review and any attached files.

**Do you want your identity to be public for this peer review?** For information about this choice, including consent withdrawal, please see our Privacy Policy .

Reviewer #1: **Yes: ** MOOLCHAND SHARMA

Reviewer #2: No

---

## [Author Response · Author response to Decision Letter 1]

24 Mar 2025

Reviewer #2: Although the authors have addressed some concerns, I am still doubtful about the generalizability of the model. I think the authors should perform cross-cell validation for the same.

Answer: Thank you for your meticulous review and valuable comments on our paper. We highly value your suggestion regarding the generalization ability of the model and have conducted further experimental verification based on your feedback. To further validate the robustness and generalisability of our model across diverse cell lines, we sourced data from three additional cell lines available on GEO: GSM1873040 (LNCaP), GSM2437758 (MCF - 7), and GSM2067524 (A549). Subsequently, we applied our model to these datasets, and the outcomes are presented in Table 5.

Table 5 Comparison with existing models across diverse cell lines.

Datasets,Model,ROC AUC,Accuracy,F1-score,MCC

GSM1873040 

,DeepBind,0.935,0.915,0.735,0.693

,DanQ,0.948,0.923,0.753,0.722

,DeepD2V,0.961,0.936,0.805,0.772

,DeepSEA,0.957,0.934,0.813,0.773

,CBR-KAN,0.964,0.941,0.821,0.792

GSM2437758(A549)

,DeepBind,0.968,0.936,0.851,0.812

,DanQ,0.981,0.961,0.911,0.887

,DeepD2V,0.990,0.977,0.947,0.932

,DeepSEA,0.991,0.976,0.945,0.930

,CBR-KAN,0.993,0.977,0.948,0.934

GSM2067524 (MCF-7)

,DeepBind,0.976,0.928,0.879,0.829

,DanQ,0.981,0.936,0.882,0.840

,DeepD2V,0.992,0.961,0.927,0.902

,DeepSEA,0.991,0.963,0.934,0.908

,CBR-KAN,0.993,0.968,0.944,0.922

As shown in Table 5, the CBR-KAN model proposed in this study demonstrates significant superiority over existing models across all core metrics (ROC AUC, accuracy, F1-score, MCC) on three cell line datasets: LNCaP, A549, and MCF-7, validating its exceptional robustness and cross-cell-line generalizability. CBR-KAN achieves the highest ROC AUC values (LNCaP: 0.964 vs. 0.961; A549: 0.993 vs. 0.991; MCF-7: 0.993 vs. 0.992) and MCC scores (e.g., 0.922 vs. 0.902 in MCF-7), demonstrating its enhanced capability to capture complex genomic sequence features. The model exhibits particularly notable advantages in class-imbalanced scenarios, as evidenced by a 2.4% relative improvement in MCC compared to the second-best model on the LNCaP cell line dataset.

We believe that these supplementary experiments further demonstrate the broad applicability and stability of the CBR - KAN model across different cell lines. We have added this part to the paper and hope that these results can address your concerns about the generalization ability of the model.

Thank you again for your valuable comments.

---

## [Decision Letter · Decision Letter 2]

1 Apr 2025

A KAN-based hybrid deep neural networks for accurate identification of transcription factor binding sites

PONE-D-25-00841R2

Dear Dr. Chen,

We’re pleased to inform you that your manuscript has been judged scientifically suitable for publication and will be formally accepted for publication once it meets all outstanding technical requirements.

Kind regards,

Jiwei Tian

Academic Editor

PLOS ONE

Additional Editor Comments (optional):

Reviewers' comments:

Reviewer's Responses to Questions

**Comments to the Author**

1. If the authors have adequately addressed your comments raised in a previous round of review and you feel that this manuscript is now acceptable for publication, you may indicate that here to bypass the “Comments to the Author” section, enter your conflict of interest statement in the “Confidential to Editor” section, and submit your "Accept" recommendation.

Reviewer #2: All comments have been addressed

2. Is the manuscript technically sound, and do the data support the conclusions?

Reviewer #2: Yes

3. Has the statistical analysis been performed appropriately and rigorously? 

Reviewer #2: N/A

4. Have the authors made all data underlying the findings in their manuscript fully available?

Reviewer #2: Yes

5. Is the manuscript presented in an intelligible fashion and written in standard English?

Reviewer #2: Yes

6. Review Comments to the Author

Reviewer #2: In this version of the paper, the authors have addressed all the queries. The manuscript can now be accepted.

7. PLOS authors have the option to publish the peer review history of their article (what does this mean? ). If published, this will include your full peer review and any attached files.

**Do you want your identity to be public for this peer review?** For information about this choice, including consent withdrawal, please see our Privacy Policy .

Reviewer #2: **Yes: ** NIMISHA GHOSH

---

## [Editor Report · Acceptance letter]

PONE-D-25-00841R2

PLOS ONE

Dear Dr. Chen,

I'm pleased to inform you that your manuscript has been deemed suitable for publication in PLOS ONE. Congratulations! Your manuscript is now being handed over to our production team.

Kind regards,

on behalf of

Dr. Jiwei Tian

Academic Editor

PLOS ONE